# Therapeutic Strategies Targeting Urokinase and Its Receptor in Cancer

**DOI:** 10.3390/cancers14030498

**Published:** 2022-01-19

**Authors:** Maria Teresa Masucci, Michele Minopoli, Gioconda Di Carluccio, Maria Letizia Motti, Maria Vincenza Carriero

**Affiliations:** 1Neoplastic Progression Unit, Istituto Nazionale Tumori IRCCS “Fondazione G. Pascale”, 80131 Naples, Italy; m.minopoli@istitutotumori.na.it (M.M.); g.dicarluccio@istitutotumori.na.it (G.D.C.); m.carriero@istitutotumori.na.it (M.V.C.); 2Department of Motor and Wellness Sciences, University “Parthenope”, 80133 Naples, Italy; motti@uniparthenope.it

**Keywords:** urokinase, urokinase receptor, urokinase inhibitors, urokinase receptor inhibitors, metastasis

## Abstract

**Simple Summary:**

The uPA/uPAR system is highly involved in cancer progression and metastasis. Many studies have definitively assessed that the high expression of urokinase-type plasminogen activator (uPA) and membrane urokinase-type plasminogen activator receptor (uPAR) in patients does correlate with metastasis formation and poor prognosis. Thus, due to the key role of uPA and uPAR in cancer, it is essential to develop compounds able to interfere with and/or inhibit their activity. In this review, we discuss the role of uPA and uPAR as diagnostic, prognostic and therapeutic markers in tumors. Moreover, we describe, in-depth, the design, construction and analysis of uPA and uPAR inhibitors in in vitro and in vivo models. Clinical trials, testing some of these inhibitors, are also accurately described.

**Abstract:**

Several studies have ascertained that uPA and uPAR do participate in tumor progression and metastasis and are involved in cell adhesion, migration, invasion and survival, as well as angiogenesis. Increased levels of uPA and uPAR in tumor tissues, stroma and biological fluids correlate with adverse clinic–pathologic features and poor patient outcomes. After binding to uPAR, uPA activates plasminogen to plasmin, a broad-spectrum matrix- and fibrin-degrading enzyme able to facilitate tumor cell invasion and dissemination to distant sites. Moreover, uPAR activated by uPA regulates most cancer cell activities by interacting with a broad range of cell membrane receptors. These findings make uPA and uPAR not only promising diagnostic and prognostic markers but also attractive targets for developing anticancer therapies. In this review, we debate the uPA/uPAR structure–function relationship as well as give an update on the molecules that interfere with or inhibit uPA/uPAR functions. Additionally, the possible clinical development of these compounds is discussed.

## 1. Introduction

Metastases are often the cause of cancer patient deaths. Unlike therapies targeting primary tumors, those targeting tumor metastases have only minimally improved. Thus, discovering more efficacious therapies able to counteract the metastatic dissemination of cancer cells is an important objective of cancer research. During the last three decades, urokinase-type plasminogen activator (uPA) and its specific receptor, uPAR, have been intensively investigated and their involvement in cancer and metastasis formation has been definitively assessed. Serine protease uPA cleaves and activates plasminogen into plasmin, an enzyme able to increase pericellular proteolysis, extracellular degradation and to improve cell invasion [1,2]. uPAR is a glycosyl-phosphatidyl-inositol (GPI)-anchored membrane receptor, comprising three-domains (DI, DII and DIII) [3]. Full-length uPAR can be cleaved by plasmin or uPA which releases the DI domain, whereas the GPI-anchored DII–DIII receptor remains either anchored to the cell surface or secreted into the extracellular milieu, in the case of the GPI-anchor cleavage [4]. Similarly, full-length uPAR or its fragments can be released in plasma and/or urine [5], as recently reviewed by Santi et al. [6]. When expressed on the cell surface, uPAR focuses uPA proteolityc activity allowing cancer cells to degrade the extracellular matrix. At the same time, uPAR is able to stimulate intracellular signaling, modulating either physiological processes as wound healing, immune responses and stem cell mobilization or pathologic events, including inflammation and tumor progression [7,8]. Upon uPA engagement, uPAR signaling occurs by assembling many different classes of molecules, composing the so-called “uPAR interactome”, comprising a total of 42 binding proteins, 9 soluble ligands and 33 lateral partners, which enables uPAR to control many pathways, regulating cell migration and proliferation [9,10]. Thus, due to the pleiotropy of its interactions, uPAR could represent a good target for designing and eventually developing compounds able to interact with uPA/uPAR and/or uPAR co-receptors. In recent years, uPA and uPAR have also been regarded as diagnostic and prognostic markers in cancer, and high levels of both uPA and uPAR in tissue and serum have been demonstrated to correlate with a poor patient outcome in different types of primary as well as metastatic malignancies [8,11,12,13,14,15,16]. In this regard, Kjaer and colleagues are developing highly specific peptide-based ligands for positron emission tomography (PET) imaging, targeting uPAR in an effort to design clinical trials for cancer patients [17]. This strategy, which employs non-invasive evaluations of primary lesions and metastatic lymph nodes, could potentially reduce the need for repeated biopsies.

In the following paragraphs, structural and functional properties of uPA, uPAR and their specific inhibitors will be accurately described, focusing the attention on peptides and small molecules which can be easily modified or engineered in order to optimize their pharmacodynamic and pharmacokinetic properties as well as their efficacy [18]. The possible clinical development of these compounds will be discussed in-depth as well.

## 2. The Urokinase Plasminogen Activator

uPA is a single-chain serine protease composed of 411 amino acid residues. The pro-enzyme (pro-uPA) is activated by the cleavage of the K158–I159 bridge by several trypsin-like proteinases, like plasmin, kallikrein, stromelysin or by glandular kallikrein mGK-6 [19,20]. The resulting fully active two-chain uPA converts plasminogen into plasmin, promoting the proteolysis of fibrinogen to fibrin and the degradation of other extracellular proteins [21]. A further cleavage at Lys 135–Lys 136 generates the amino-terminal fragment (ATF, amino acids 1–135), which includes an EGF-like domain, i.e., the growth factor domain (GFD) (residues 1–49 of the uPA human sequence) and the kringle (amino acids 50–131) domain. The remaining 136–158 fragment, called the connecting peptide (CP), is linked through a disulfide bridge to the carboxy-terminal catalytic region in the low molecular weight form (LMW uPA, amino acids 135–411), which retains the catalytic activity [21,22,23,24,25]. Notably, these multi-domain structures can also act as independent folded compact modules [26]. Three binding regions have been identified to interact with uPAR: the first one, contributing to the high-affinity uPA/uPAR binding, is constituted by the ATF hydrophobic residues Phe25, Ile28 and Trp30 which interacts with the DI uPAR domain; the second region, formed mainly by one stretch of residues in the GFD of uPA (Ser21, Asn22, Lys23 and Tyr24), contacts mainly the DII uPAR domain; the third region involves hydrogen bonds and van der Waals contacts between ATF residues and the DI uPAR domain [27,28]. Of note, besides full length and membrane-associated uPAR, uPA also binds soluble forms of uPAR (SuPAR) secreted in the extracellular milieu [29].

Pro-uPA, uPA and ATF bind to uPAR at a similar extent with a Kd within a 100–500 pM range, whereas the GFD or its shorter peptide, corresponding to residues 18–32, bind to uPAR with a 1000-fold lower affinity, being the Lys23, Tyr24, Phe25, IIe28 and Trp30 residues essential for uPAR binding [23,27]. Inhibitors of uPA have been constructed, interfering with either uPA catalytic activity or uPA-dependent signaling.

## 3. The Urokinase Plasminogen Activator Receptor 

The urokinase receptor is a GPI-anchored membrane receptor, first identified on blood monocytes and the U937 monocyte-like cell line [30,31]. By binding to uPA at the leading edge of migrating cells, uPAR orchestrates extracellular proteolysis and cell invasiveness in the near tissues [32]. The mature uPAR moiety is a cysteine-rich molecule constituted of three domains (DI, DII and DIII from the N-terminus) connected by linker regions and associated with the cell membrane through a C-terminal GPI anchor which is characterized by intra-chain disulfide bonds [33]. The DI domain contains the uPA-binding site, albeit the entire uPAR molecule is required for an efficient binding [34,35]. The DII and DIII uPAR domains are also involved in the binding, increasing the DI domain affinity for uPA [36]. 

uPAR can be cleaved in the DI–DII linker region not only by uPA but also by several proteases, like trypsin, chymotrypsin, elastase, cathepsin G, metalloproteases and plasmin [37]. By proteolysis, uPAR releases the DI binding domain, whereas the truncated GPI-uPAR remains at the cell surface and exposes the N-terminus uPAR84-95 chemotactic sequence [38]. Moreover, by the proteolysis of the GPI-anchor, soluble forms of uPAR are produced and poured into the extracellular milieu and bloodstream. The presence of high levels of uPAR in serum indicate pathological conditions, including inflammation and cancer [39]. The crystal structure of uPAR bound to a competitive peptide inhibitor of the uPA/uPAR interaction demonstrated the occurrence of three-finger consecutive domains generating a deep cavity where the peptide binds and a large external surface that remains accessible for protein–protein interactions [34,35]. By analyzing the conformational preferences of uPAR, Yuan and Huang demonstrated that uPAR has a latent inactive form activable by uPA-induced conformational changes [40]. Gardsvoll et al. assessed that uPAR shows a high conformational flexibility. After uPA binding, uPAR shifts from an open to an intermediated and then to a closed conformation [41], the last being able to form potent multiprotein cell-signaling complexes which regulate migration, invasion, metastasis, epithelial–mesenchymal transition, stem cell-like properties, survival, release from states of dormancy, chemo-resistance, angiogenesis and vasculogenic mimicry in cancer [10,42,43,44,45,46,47] (Figure 1).

uPAR regulates cytoskeletal rearrangements, cell adhesion and migration by a direct binding to the somatomedin-like domain of vitronectin (Vn), and uPA engagement increases the uPAR binding affinity for Vn [48,49,50]. The association between uPAR and Vn is mediated by the DI–DII interface through the residues Trp32, Arg58, Ile63 (in DI), Arg91 and Tyr92 [48,51,52,53]. The residues Arg91 and Tyr92 in the DI–DII linker region have been shown as essential for the interaction of somatomedin B and Vn [53] and crucial for increasing Vn-induced, uPAR-mediated cytoskeletal rearrangements and migration [52]. A number of studies document that uPAR starts or increases the signaling induced by integrins through focal adhesion kinase and/or Src kinases by activating the MEK/ERK pathway [44,54,55,56]. uPAR interacts with the αvβ5 or αvβ3 Vn receptors (VnR)s, promoting cytoskeleton reorganization, cell adhesion and migration [57,58,59]. The connection between the uPA and uPAR system and VnRs is also highlighted by the fact that uPA or its 136–158 connecting peptide directly interacts with the αv chain of VnR, thus, creating a uPA-mediated bridging of the uPAR and αv chain of VnR [60]. The association of uPAR with integrin α3β1, which recognizes several ECM ligands, including laminins, type IV collagen and fibronectin, has been reported to occur via a surface loop within the β-propeller of the α3 subunit but outside the laminin binding region [61]. The α5β1integrin/uPAR interaction supports cell adhesion to fibronectin, migration toward fibronectin and fibronectin matrix assembly [62,63,64]. uPAR itself promotes the EGFR interaction with the integrin α5β1, leading to the formation of uPAR/EGFR/α5β1 complexes which, recruits FAK and activates the mitogenic Raf/MEK/ERK signaling pathway [65,66]. Interestingly, the inhibition of uPAR association with β1integrin, or EGFR alone, induces cell dormancy, thereby resulting in tumor suppression [55,65,67]. Both DII and DIII domains participate in the interaction of uPAR with integrins. In the uPAR DII domain, Degryse et al. identified the sequence D2A (30IQEGEEGRPKDDR142) which directly interacts with the αvβ3 VnR and stimulates cell migration by activating αvβ3-dependent signaling pathways, including the Janus kinase/Stat pathway [68]. Of note, in addition to chemotactic activity, the D2A-derived peptide has been shown to promote cell growth by trans-activating EGFR [69]. Moreover, Chaurasia et al. found that the sequence 240GCATASMCQ248 located in the DIII uPAR domain is implicated in the lateral interaction with the α5β1 integrin, leading to ERK pathway activation and tumor growth [70]. Formyl-peptide receptors type 1 (FPR1) and 2 (FPR2) are members of the seven-transmembrane, G-protein-coupled receptors superfamily, which are expressed on polymorphonuclear and mononuclear phagocytes as well as on non-hematopoietic cells including endothelial cells and fibroblasts [71,72]. It has been assessed that FPR1 is involved in the progression of solid tumors including glioblastoma, neuroblastoma, sarcoma, melanoma and ovarian cancer [73,74,75,76,77]. The binding of agonists to FPR1 promotes a cascade of signals, involving PI3K, MAPK and the transcription nuclear factor (NF)-κB, eliciting several responses such as morphological polarization, chemotaxis, the production of reactive oxygen species and release of cytokines and proteolytic enzymes [78]. Upon uPA engagement, the change in uPAR conformation unmasks the uPAR84–95 sequence situated in the region between the domains DI and DII, triggering chemotactic activity by interacting with FPR1 or FPR2 [79,80,81,82,83,84]. Upon binding to FPR1, the uPAR84–95 sequence or a shorter uPAR88–92 sequence corresponding to the synthetic, linear peptide Ser-Arg-Ser-Arg-Tyr (SRSRY), bind to FPR1, cause FPR1 internalization and trigger VnR activation with an inside-out type of mechanism, resulting in increased cell migration and angiogenesis [71,82]. It should be noted that uPAR controls angiogenesis either by focusing surface-associated proteolytic activity [85,86,87] or in an uPA-independent manner. Rao JS et al. demonstrated that SuPAR released by cancer cells in the extracellular milieu may be recruited onto lipid rafts of umbilical vein endothelial cells (HVECs), thus, promoting Rac1-mediated endothelial cell migration and tumor angiogenesis [88]. uPAR has been demonstrated to trigger integrin redistribution, by which VEGF controls endothelial cell migration [89]. However, uPAR itself may interact with VEGFR2 via the low-density lipoprotein receptor-related protein 1 (LRP-1) which, in turn, causes the VEGF–VEGFR2 internalization, necessary for angiogenesis [90]. More recently, Chillà A and coworkers described a new type of tumor neovascularization, the “amoeboid angiogenesis”, which depends on uPAR/integrin αvβ3 interaction. This amoeboid movement due to a protease-independent amoeboid migration of endothelial cells is VEGF-independent, is faster than the protease-dependent mesenchymal migration and may be controlled by a 25mer peptide that inhibits uPAR/integrin contacts [91].

It has been shown that either uPAR and DII–DIII-SuPAR regulate CXCR4 through FPR1 [83,92], and that the expression of uPAR and CXCR4 is modulated by microRNAs (miR)s in acute myeloid leukemia cells. Indeed, miR-146a and miR-335 directly target the 3′UTR of uPAR and CXCR4, thus, interfering with the expression of both uPAR and CXCR4 mRNAs [93]. By the activation of the transcription of Bcl-xL, uPAR can also promote cell survival via the MEK/ERK and PI3K/Akt-dependent pathways [94]. 

Due to its multiple interactions with numerous ligands, uPAR is a very interesting target for designing therapeutic cancer drugs inhibiting uPAR functions. Therefore, besides uPA inhibitors, different inhibitors of uPAR functions have been designed. Among them, inhibitors of the interaction between uPAR and integrins, G protein-coupled receptors and EGFR have been constructed and analyzed.

## 4. Inhibitors of uPA 

### 4.1. Inhibitors of uPA Catalytic Activity

Among the inhibitors interfering with uPA catalytic activity, benzylsulfonyl-D-Ser-Ser 4-aminobemzylamide has been shown to exert strong anti-metastatic activity in the lungs of a mouse model with fibro-sarcoma. This compound was well-tolerated and showed no side effects when administrated in mice, either intraperitoneally or subcutaneously [95]. WX-UK1 (Table 1), a derivative of 3-aminophenylalanine, was described by Setyono-Han and coworkers. In the L-enantiomer form, WX-UK1 added at nano-molar concentrations to breast cancer cells, inhibited the proteolytic activity of uPA, plasmin and thrombin in a dose-dependent manner. Accordingly, WX-UK1 exerted anti-tumor and anti-metastatic activity in a breast tumor rat model, without eliciting side effects [96]. A pro drug of WX-UK1, i.e., WX-671 (Mesupron) was investigated to treat metastatic breast cancer and pharmacokinetic studies have been accomplished [97,98]. Goldstein LJ described a set of clinical trials with WX-671 or upamostat (or Mesupron) (Table 1). 

Mesupron was tested in monotherapy, either in a Phase I study, to investigate the bioavailability, tolerability and pharmacokinetic profile in healthy male subjects, at single intravenous increasing doses (WX/50-001), or in a Phase I/IIa monotherapy study in patients with advanced solid tumors (WX/50-003) or in a Phase Ib monotherapy study in advanced head and neck tumors (WX/50-004). Finally, a Phase I combination study with capecitabine in patients with advanced solid tumors was accomplished (WX/50-005). Additionally, the bioavailability, tolerability and pharmacokinetic profile of orally administrated WX-671 was studied in healthy males (WX60/-001 and WX60/-002) and in patients with head and neck carcinomas (WX/60-003). The combination of WX-671 per os, plus capecitabine compared to capecitabine alone, was investigated in a phase II study (WX60/-006) in patients with HER2-negative metastatic breast cancer, following first-line chemotherapy [97]. It has been demonstrated that mesupron plus auranofin, a thioredoxin reductase inhibitor, exerted a significant inhibition of breast cancer cells via apoptosis and the arrest of cells in the G1/S phase [99]. In patients with non-resettable, locally advanced pancreatic cancer, a Phase II randomized clinical trial of upamostat in combination with gemcitabine versus gemcitabine alone was sponsored by Heidelberg Pharma AG with the aim to assess the efficacy and tolerability. Starting from 2007, 95 patients were enrolled and centrally randomized in a 1:1:1 ratio and received the same dose and schedule of gemcitabine alone or in addition with a daily oral dose of either 200 mg or 400 mg upamostat. In all the evaluated subjects, the combination was safe and well-tolerated, with the adverse events of upamostat being asthenia, fever and nausea. In 12 patients, progression to tumor metastasis was demonstrated [100]. 

The [3-(4-chlorophenyl)-adamantane-1-carboxylic acid (pyridin-4-ylmethyl)amide], named ABC294640 or opaganib, is a sphingosine kinase-2 (SK2) selective inhibitor with anticancer activity against multiple cancer-type lesions [101,102]. On 20 October 2017, FDA granted the use of the mesupron orphan drug for the adjuvant treatment of pancreatic cancer. In November 2020, RedHill Biopharma Ltd. received a U.S. patent allowance covering the opaganib and RHB-107 (mesupron) combination for treatment of solid tumors. The combination of opaganib and RHB-107 causes the regression of cholangiocarcinoma patient-derived xenografts. Data were illustrated at the American Association for Cancer Research (AACR) in 2020 [103]. By screening a library of chemical compounds, Zhu M. and collaborators identified several inhibitors of uPA catalytic activity. The study pointed out 4-oxazolidinone as a novel lead pharmacophore, whose potency and selectivity were enhanced by structural modifications. The analogue, UK122, showed a strong inhibition of uPA activity in vitro. Moreover, UK122 inhibited the migration and invasion of pancreatic adenocarcinoma cells eliciting low cytotoxicity [104].

### 4.2. Peptide Inhibitors of uPA-Binding to uPAR, uPA Proteolytic Activity and uPA-Dependent Signaling Activity

By chemical modifications of the GFD, many compounds able to interfere with or antagonize uPA binding to uPAR and inhibit uPA proteolytic activity at the cell surface have been designed. Among these, the cyclo19,31[D-Cys19]-uPA19-31 was suggested as a promising possible drug in uPAR overexpressing tumor cells for its ability to inhibit uPA/uPAR interaction, disrupting uPA binding to uPAR and abrogating uPA plasminogen activation at the cell surface and fibrin degradation [105]. Sato and co-authors described two small cyclic peptides able to prevent the uPA/uPAR interaction: WX-360 (cyclo(21-29)[D-Cys21]-uPA(21-30)[S21C;H29C]) and its norleucine (Nle) derivative WX-360-Nle(cyclo(21-29)[D-Cys21]-uPA(21-30)[S21C;K23Nle;H29C]). In nude mice, WX-360 and WX-360-Nle suppressed the growth and intraperitoneal spread of lacZ-tagged human ovarian cancer cells [106]. By using combinatorial chemistry, Ploug et al. designed and analyzed a 9-mer, linear peptide (named AE105) able to antagonize the uPA/uPAR interaction, and demonstrated its specificity and high-affinity for human uPAR (Kd ~ 0.4 nM) [107]. In functional assays, the specific inhibition of cancer cell intravasation was demonstrated by inoculating HEp-3 cancer cells in a chicken chorioallantoic membrane in the presence of AE105 [107]. Derivatives of this peptide were instrumental for testing probes to image and/or target uPAR in patients with solid cancers [108]. The therapeutic efficacy was demonstrated by inoculating a DOTA-conjugated AE105 peptide (DOTA-AE105) in a human xenograft colorectal cancer model. Either tumor cell growth or uPAR cell content in the tumor tissue were reduced by this treatment [107,109]. Several radio-ligands based on AE105 were developed for PET imaging and targeted radionuclide therapy [110,111,112,113]. Phase 1 studies with both 68Ga-NOTA-AE105 and 64Cu-DOTA-AE105 (Table 1) in patients with various cancer types have been published [16,114]. Both ligands accumulated in primary tumor lesions and in metastases, and the uptake corresponded to high uPAR expression detected in the excised tumor tissues; these data provided evidence for highly targeted specific uPAR PET imaging. Phase 2 studies in which 68Ga-NOTA-AE105 is injected as PET tracer are summarized in Table 1.

Other inhibitors, derived from uPA regions not involved in the binding to uPAR have been designed and analyzed for their ability to inhibit tumor progression. The recombinant kringle region spanning from Asp45 to Lys135 of uPA (UK1: Asp(45)-Lys(135) was found to inhibit the proliferation and migration of endothelial cells depending on VEGF and uPA as well as angiogenesis on the chick chorioallantoic membrane [115]. Systemic administration of UK1 inhibited the growth of U87 human glioma cells implanted into nude mice brains. Immuno-histochemical analysis of UK1-treated tumors revealed a reduced vascularization and lower expression of angiogenesis-related factors, including VEGF, angiogenin, α smooth muscle actin, von Willebrand’s factor and the platelet endothelial cell adhesion molecule-1, compared to untreated mice, and increased apoptosis [116]. In malignant gliomas, the same authors studied the combined use of UK1 and celecoxib, an inhibitor of cyclooxygenase type 2, able to suppress the growth of tumor tissues and vessels. In vitro, the combined treatment of UK1 plus celecoxib increased endothelial cell proliferation, migration and tube formation inhibition, depending on UK alone, but did not prevent U87 cell proliferation. Moreover, in nude mice, on a U87 glioma cells xenograft, the combined therapy of UK1 and celecoxib elicited a higher inhibition of tumor growth compared to monotherapy, increasing apoptosis, and decreasing the levels of pro-angiogenic factors. In endothelial cells, UK1 reduced VEGF and the phosphorylation of ERK1/2 depending on bFGF action. Thus, the synergistic therapy with UK1 and celecoxib could be a good therapeutic option for gliomas [117]. 

The 8-mer capped peptide corresponding to the 135–143CP residues (Acetyl-Lys-Pro-Ser-Ser-Pro-Pro-Glu-Glu-NH2), named Å6, inhibited the binding of uPAR and SuPAR to uPA in a dose-dependent manner at concentrations of Å6 as low as 0.25 µM. In vitro, Å6, inhibited the invasion of breast cancer cells and the migration of endothelial cells, in a dose-dependent manner, with an IC50 between 5–25 µM. Intraperitoneal, continuous infusion or twice daily administration of 75 mg × kg-1 Å6 promoted an evident inhibition of tumor growth, causing a decrease in micro-vessel density and lymph node metastasis, leading to extensive tumor necrosis [118]. Combination treatments with Å6 and the antiestrogen tamoxifen were tested on estrogen-receptor-positive Mat B-III rat breast cancer cells, in vitro and in vivo. In vitro, the combination of Å6 plus tamoxifen elicited a higher dose-dependent decrease in tumor-cell invasion, compared to monotherapy. In orthotopic rat models, the combined administration of Å6 and tamoxifen caused a 75% reduction in tumor growth, in comparison to control animals and an anti-metastatic effect, the last being due exclusively to Å6. In samples from animals submitted to the combination therapy, the reduction in blood vessels and the rise in tumor cell death were evidenced by histological analysis. These findings allowed the authors to conclude that the addition to hormone therapy of Å6 could enhance the antitumor effects of tamoxifen [119]. Similar encouraging results were obtained on glioblastoma models. In vitro, K Mishima et al. demonstrated that Å6 inhibits human endothelial cell migration without affecting proliferation of either human endothelial or U87MG glioma cells. In vivo, Å6 and cisplatin inhibited subcutaneous tumor growth by 48% and 53%, respectively, whereas the combined treatment of Å6 plus cisplatin reduced tumor growth by 92%, increasing animal survival and reducing intracranial tumor size and neovascularization in xenografts [120]. After the good results obtained either in vitro or in animal models, the uPA Å6-derived peptide has also been studied in clinical trials (Table 1). A clinical phase I trial for Å6 was accomplished in patients with advanced gynecologic cancers. The study enrolled 16 patients. Daily subcutaneous administration of Å6 did not cause side effects, apart from local reactions at the injection site. This study showed that 300 mg Å6/daily continuously administrated is well-tolerated and that 5 out of the 16 treated patients had no increase in tumor size [121]. Based on the phase I trial results, phase II studies were accomplished in order to assess the clinical efficacy and safety of Å6 (Table 1). Patients with epithelial ovarian, fallopian tube or primary peritoneal cancer in clinical remission after first-line chemotherapy and with two consecutive increases in CA125 but no clinically or radiologically demonstrable disease were divided in three groups and received either placebo, low-dose or high-dose Å6 until disease progression or exit from the study. Primary endpoints were time to clinical disease progression and safety of Å6. Evaluation of changes of CA125 levels and biomarkers related to the uPA/uPAR system in serum were the secondary endpoints of the study. Analysis of the results concluded that Å6 therapy is well-tolerated and extends the time to clinical disease progression, whereas it does not elicit any CA125 response [122]. The activity and tolerability of Å6 in patients with persistent or recurrent epithelial ovarian, fallopian tube or primary peritoneal carcinoma were evaluated in another study. Å6 was well-tolerated but exerted such side effects [123]. These contradictory results can be explained by the different clinical characteristics of the patient populations enrolled in the two studies. Phase II trials of Å6 were accomplished not only in ovarian cancer but also in patients with chronic lymphocytic leukemia and small lymphocytic lymphoma to assess the efficacy, safety and pharmacodynamic markers in these hematological pathologies (lead sponsor Ångstrom Pharmaceuticals, 2014–2016) (Table 1).

Other molecules interfering with uPA activity have been designed based on different criteria. Franco et al. described the structural features and the biological activity of two adjacent regions in the CP region of uPA, affecting cell migration in an opposite way. The study was based on several synthetic peptides, including the N- and C-capped uPA-(135–143) 9-mer and the N-acetylated uPA-(144–158) 15-mer, corresponding to the N-terminal [uPA-(135–143), residues 135–143 and the C-terminal [uPA-(144–158) and residues 144–158 of the CP region. These peptides bind uPAR and are able to displace ^125^I-CPp, though at different affinities. The uPA-(144–158) peptide showed chemotactic activity for embryonic kidney HEK293/uPAR-25 cells, favoring migration in Boyden chambers, whereas the uPA-(135–143) peptide inhibited serum- and CP peptide-induced cell migration [124]. More recently, two new decapeptides, corresponding to the residues 136–145 in the CP region of human uPA, named Pep 1 and Pep 2 (i.e., the cyclic derivative of Pep 1), were designed and analyzed for their functional effects. Both peptides inhibit cell migration and invasion, in vitro and in vivo, and the dissemination of fibrosarcoma and mammary carcinoma cells implanted in nude mice. In nude mice, Pep 2 also reduced the number and size of HT1080 fibrosarcoma lung metastases. In 3D-organotypic co-cultures, both peptides reduced HT1080 fibrosarcoma and MDA–MB-231 mammary carcinoma cell invasion by inhibiting the pro-invasive activity of telomerase immortalized fibroblasts (TIFs) and primary cancer-associated fibroblasts (CAF)s derived from breast carcinoma patients. Pep 1 and Pep 2 bind specifically to the αv integrin subunit, and TIFs ability to chemoattract cancer cells and contract collagen matrices is reduced by αv integrin silencing. Interestingly, TIFs or primary CAFs exposure to these peptides reduced α-smooth muscle actin levels downregulating their matrix contracting ability [125]. Due to the clear-cut effects of Pep 1 and Pep 2 on tumor cell invasion, the possibility of their use as therapeutic lead compounds should be investigated in-depth.

## 5. Inhibitors of uPAR 

### 5.1. Peptide Inhibitors of the uPAR/Integrin Interaction

Simon and collaborators identified a critical non-I-domain binding site for uPAR on CD11b (residues 424PRYQHIGLVAMFRQNTG440). The corresponding peptide named M25 interrupts uPAR association with beta1 and beta2 integrins, impairing integrin-dependent spreading and migration of human vascular smooth muscle cells on fibronectin and collagen [126]. The uPAR/integrin complex can be inhibited by a 17-amino acid peptide (P25), isolated from a phage peptide library [127]. The P25 peptide, which is homologous to M25, decreased the adhesion of MDA-MB-231 breast cancer cells to Vn and increased their adhesion onto fibronectin [128]. Wei et al. demonstrated that the 17-mer α3 peptide α325 (residues 241–257 of the α3 integrin chain) blocks uPAR-dependent cell adhesion onto Vn, this effect being nearly identical to that exerted either by M25 and P25 [129]. Supurna Ghosh et al. found that the peptide α325 inhibits α3β1/uPAR binding, reducing the integrin-correlated uPA overexpression in oral squamous cell carcinoma cells [130,131]. Moreover, starting from the 224–232 sequence in the β1-chain, which is involved in the uPAR/α5β1 physical association, Ying Wei and colleagues designed the peptide β1P1 carrying the Ser227 substituted with an alanine residue. β1P1 totally abolished the uPAR/α5β1 interaction [64]. By substituting two alanine with two glutamic acid residues in the D2A chemotactic sequence, Degryse and collaborators constructed the analogue D2A-Ala which inhibits and disrupts the uPAR/αvβ3 and uPAR/α5β1 complexes, i.e., the signaling. Both D2A-Ala and the shorter derived tetrapeptide GAAG are not chemotactic, do not exert any signaling activity but behave as powerful integrin-dependent cell migration inhibitors [68].

### 5.2. Peptide Inhibitors of uPAR/FPR1 Interaction

Some years ago, we found that the residue Ser90, located in the chemotactic uPAR84–95 sequence influences the conformation of the nearby residues and that the substitution of Ser90 with a glutamic acid residue in the membrane-associated uPAR, prevents agonist-triggered FPR1 activation and internalization, and interferes with uPAR/FPR1/VnR crosstalk, blocking cell migration and invasion, both in vitro and in vivo [132]. Following this observation, two linear peptides (pGlu)-Arg-Glu-Arg-Tyr-NH2, named pERERY, and N-terminal acetylated, C-terminal amidated Ac-Arg-Glu-Arg-Phe-NH2, named RERF, were selected for their ability to inhibit the uPAR/FPR1 interaction, reducing directional cell migration, invasion and angiogenesis to basal levels [84,133,134]. To overcome their instability to enzymatic digestion in human serum, the synthesis of other compounds was accomplished. By cyclization of the chemotactic SRSRY linear peptide, a new inhibitor of the uPAR/FPR1 interaction, named [SRSRY], was generated [135]. [SRSRY] inhibits FPR1-mediated cell migration and prevents both the internalization and ligand-uptake of FPR1 in rat basophilic leukemia RBL-2H3 cells which constitutively express human FPR1. [SRSRY] was found to inhibit cell migration in a dose-dependent manner, with an IC50 value of 0.01 nM, and to prevent the trans-endothelial migration of monocytes [135,136]. Moreover, by applying a recto-verso approach, which ensures resistance to enzymatic digestion [137], a new library of RERF-derived peptides was developed. Among these, the retro-inverso peptide Ac-(D)-Tyr-(D)-Arg-Aib-(D)-Arg-NH2 (named RI-3) results to be the best inhibitor of cell migration mediated by uPAR via FPR1, with an IC50 value of 10 pM [75]. RI-3 inhibits the migration, invasion and trans-endothelial migration of sarcoma cells as well as angiogenesis and lung metastasis [75]. In nude mice injected subcutaneously with sarcoma cells, RI-3 reduced tumor size and micro-vessel density as well as circulating tumor cells [75]. Ragone et al. extensively analyzed the functional relationship between RI-3 and FPR1 in A375 and M14 melanoma cells: RI-3 fully prevented the invasion of both melanoma cell lines, either in Boyden chambers or in 3-D organotypic models [76]. Upon exposure to 10 nM RI-3, epithelial ovarian cancer cell adhesion onto mesothelial cell monolayers and mesothelial invasion were inhibited, suggesting that RI-3 may be useful for blocking the intra-abdominal dissemination of ovarian cancer cells [77]. Recently, Minopoli et al. demonstrated a high reduction in monocytes recruitment and infiltration into tumor tissues by treatment with 6 mg/Kg RI-3/die nude mice xenografted with primary chondrosarcoma cells [138]. These results indicate the chance that RI-3 could be developed as a drug for adjuvant therapy of tumors expressing uPAR and FPR1.

## 6. Small Molecules Affecting the uPA/uPAR System

β-Elemene, a phytochemical compound from a Chinese herb showing anticancer activity against leukemia and some solid tumors, has been documented to inhibit tumor growth by downregulating uPA, uPAR, MMP-2, and MMP-9 expression at mRNA and protein levels in a murine intraocular melanoma model [139]. In the past years, by a virtual screening approach, small molecules able to inhibit the interaction between uPA and uPAR have been identified by competition and biochemical assays [140]. Amiloride-HCl, which is an oral potassium-sparing diuretic, has been shown to exert anti-tumor and anti-metastasis activities through uPA and the sodium–hydrogen exchanger1 (NHE1), a membrane protein responsible for the low extracellular pH in tumors [141]. Buckley and co-workers screened substituted amiloride libraries in order to identify amiloride-derivatives that selectively inhibit uPA or NHE1 and identified a pyrimidine-substituted 5-(*N*,*N*-hexamethylene)amiloride (HMA) analog showing significant activity (IC50 < 300 nM) on both uPA and NHE1 targets and minimal effects on cell viability [142]. By structure-based computational studies, IPR-456 was found to bind uPAR with a sub-micromolar affinity and to block its interaction with ATF with an IC50 of 10 µM. In vitro, IPR-456 inhibited the invasiveness of breast cancer cells and, in mice, the formation of lung metastasis. Cell adhesion, migration and invasion of non-small lung cancer cell lines were also hampered [143,144]. Small molecules inhibiting with high affinity uPAR binding to uPA were constructed. They make uPAR unable to bind uPA, modifying its conformation [145,146]. Using a biophysical approach, molecular docking and extensive explicit-solvent molecular dynamics simulations of uPAR bound to uPA, Xu, D. et al. demonstrated that the small compound IPR-3011 blocks uPA binding to both open and closed conformations of uPAR [146]. 

By structure-based virtual screening, Rea and co-workers searched small molecules targeting the uPAR-binding site for Vn. They identified the C6 and C37 compounds targeting the residues S88 and R91, which are documented to be located in the binding region of uPAR for Vn and are involved in uPAR binding either to VN and FPRs. C6 and C37 compounds inhibited cell migration and ECM invasion by several cancer cell-types in vitro, and can be regarded as promising pharmaceutical lead compounds for cancer treatment [147]. Small molecule drugs may represent an alternative approach to interfere with uPAR interactome. Two molecules able to disrupt the uPAR/alpha5beta1integrin association have been identified, named 2-(Pyridin-2-ylamino)-quinolin-8-ol and 2,2′-(methylimino) di (8-quinolinol). They cause tumor cell dormancy and are able to inhibit ERK activation, tumor growth and metastasis in a model of head and neck carcinoma in vivo [148]. Finally, Nagarekha Pasupuleti et al. described the 5-Benzylglycinyl-amiloride (UCD38B), an anticancer molecule showing cytotoxic effects both on proliferative and non-proliferative high-grade glioma cells. UCD38B makes a relocation to perinuclear mitochondrial regions in 40–50% of endosomes containing uPA, uPAR and the uPA inhibitor PAI-1. UCD38B induced endosomal “mis-trafficking” in human glioma cells, causing mitochondrial depolarization, the release and nuclear translocation of the apoptosis-inducing factor and consequent cell death [149].

## 7. Genetic Inhibitors of uPA and uPAR Expression

Antisense RNA technology has been successfully used to downregulate uPAR expression in tumors, either in vitro or in murine models, using both plasmid and adenovirus constructs [150]. Firstly, Quattrone et al. described an anti-messenger uPAR oligodeoxynucleotide sequence (uPAR aODN, 5′-CGG-CGG GTG ACC CAT GTC-3′) able to switch off uPAR gene expression and to abolish invasiveness of SV40-transformed human fibroblasts, providing a possible therapy to be tested in in vivo studies [151]. Later, the administration of uPAR aODN was found to reduce bone metastasis formation by PC3 human prostate cancer cells in a mouse model. Three groups of nude mice were intraventricular injected with PC3 cells and intraperitoneally treated either with uPAR aODN, a degenerated ODN or saline solution. Margheri et al. found that only 20% of the mice treated with uPAR aODN developed metastases, as compared to controls [152]. Accordingly, an 80% reduction in tumor size and the absence of lung metastases was demonstrated in mice inoculated with osteosarcoma cells transfected with an antisense uPAR vector [153]. Go Yi et al. showed that human glioblastoma cells transfected with a cDNA construct corresponding to 300 bp of the human uPAR 5′ did not cause intracerebral tumor lesion after injection in nude mice [154]. Tavian D et al., using an antisense vector containing the 5′ portion (257 bp) u-PA cDNA, found that the stable expression of antisense urokinase mRNA inhibits the proliferation and invasion of human hepatocellular carcinoma cells [155]. Wu X et al. demonstrated that a COX-2 antisense ODNs is able to inhibit the invasiveness of OS-732 human osteosarcoma cells in vitro, by decreasing both mRNA and protein levels of COX-2, uPA and uPAR [156]. An 80–90% reduction in uPAR levels in invasive human lung cancer cells was seen by an antisense strategy using an adenovirus construct (Ad-uPAR). These modified cells showed a 70% decrease in matrigel invasion. Moreover, the infection of tumor cells with Ad-uPAR before implantation into nude mice reduced the incidence of lung metastasis by 85% [157]. By cell infection with a bicistronic construct containing antisense sequences of uPAR and uPA, neither tumorigenity nor the invasion of a glioma cells injected intracranially in nude mice were found. Moreover, this construct determined the regression of established tumors after administration in vivo [158]. Finally, the intra-peritoneal injection of a shRNA expressing plasmid targeting either uPAR and uPA caused tumor regression in mice bearing intracranial gliomas [159]. Alfano et al. identified miR-146a, miR-335 and miR-622 in leukemia cells that directly target the 3′untranslated region of both uPAR- and CXCR4-mRNAs, thus, downregulating both uPAR and CXCR4 expression [93].

Very recently, on human endothelial cells, Biagioni et al. have demonstrated the pro-angiogenic effects of uPAR containing esosomes derived from melanoma cell lines. By CRISPR–Cas 9, an innovative technology leading to the permanent suppression of gene expression, they obtained a significative uPAR knockout, resulting in a high reduction in uPAR pro-angiogenic effects by the inhibition of VE-Cadherin, EGFR and uPAR expression and ERK1,2 signaling in human endothelial cells, both in vitro and in vivo [160]. Taken together, these findings strongly support the possibility to counteract tumor progression by using genetic therapies devoted to inhibiting uPA and uPAR at mRNA and protein levels. 

## 8. Bioengineered Drugs Affecting the uPA/uPAR System

New possible therapeutic strategies have been investigated, using bioengineered drugs generated by the recombinant DNA technology, i.e., molecules characterized by a high and specific activity in the presence of optimal safety. These bioengineered molecules, named ligand-targeted toxins (LTT)s, are constituted of components which bind specific targets and carry toxin inducing cell death. Therefore, they cause death only of the cells able to specifically bind them. Moreover, by combining agents targeting the uPA system with nanobins, it is possible to target precisely tumor cells [161], deliver a larger amount of cytotoxic drugs in the tumor cells [162] and increase drug circulation half-life and specific cytotoxicity [163].

ATF conjugated to the diphtheria toxin (DTAT) caused the decrease in glioblastoma tumor growth [164,165]. In a mouse model of human metastatic non-small cell lung cancer cells (NSCLC) to the brain, Huang J et al. investigated the cytotoxic effect of the bispecific diphtheria toxin-based immunotoxin (DTATEGF), which targets both EGFR and uPAR. DTATEGF killed NSCLC cells in vitro and in vivo in a human metastatic NSCLC intracranial model. Overall survival of treated mice was longer compared to that of the untreated animals [166]. ATF has been combined to nanoparticles in order to target uPAR. Yang et al. conjugated iron oxide (IO) nanoparticles to ATF and demonstrated a specific binding to uPAR expressing breast cancer cells, followed by internalization of the nanobins. In mice model, ATF-IO do accumulate in mammary tumors and can be used as a double target tracer for breast cancer imaging (MRI) [167]. Abdalla et al. have constructed theranostic ATF conjugates containing IO and noscapine, cytotoxic for prostate cancer cells in a μM range [168]. Photodynamic therapy selectively destroys malignant cells while sparing the normal tissues and, thus, is recognized as a minimally invasive and toxic treatment strategy. In this regard, Chen et al. described a series of uPAR-targeting anticancer agents, constructed by covalently binding zinc phthalocyanine (ZnPc) to the amino-terminal fragment (ATF) of uPA, to be used for photodynamic therapy. ATF-ZnPc competed with ATF for uPAR binding, with an IC50 = 8.6 nM and enhanced the antitumor specificity and efficacy of ZnPc in vitro and in vivo on different cancer cell-types [169]. Recently, the same authors constructed nanoparticles endowed with a larger amount of ZnPc photosensitizer, eliciting a higher photodynamic effect [170]. A fusion protein constituted of ATF and human serum albumin (ATF-HSA) has been shown to facilitate drug delivery ensuring a longer circulating time and higher drug carrier properties. Thus, it can be used either as a drug or as an agent for imaging of uPAR-expressing tumors [171]. When the hydrophobic photosensitizer (mono-substituted β-carboxy phthalocyanine zinc, CPZ) was used as a cytotoxic agent and embedded inside ATF-HSA, the resulting ATF-HSA:CPZ retained uPAR-targeting binding ability coupled to high stability and optical and photo-physical properties. Moreover, ATF-HSA:CPZ was proven to accumulate specifically in tumors and exhibit tumor-killing ability in a mouse model at a dose of 80 nmol/kg of mouse body weight [172]. ATF-HSA-loaded doxorubicin (DOX) was constructed, having a higher antitumor effect compared to free DOX while showing a reduced cardiotoxicity [173]. 

A very recent and detailed review by Felix Oh and colleagues discuss the possible therapeutic options of bispecific LTTs constructed by conjugating cytotoxic drugs to components targeting uPAR and EGFR. These molecules counteract tumor growth and intra-tumor micro-vessel density in a variety of solid tumors, also interfering with the pro-tumor activities of uPAR expressing tumor-associated macrophages [174]. Firstly, Vallera and co-workers constructed a bispecific ligand-directed toxin (BLT) which recognizes two receptors overexpressed on the same cell [175]. Based on these findings, the same research group developed and tested a bispecific LTT targeting either uPAR and EGFR, thus, enhancing both the specificity of targeting and the toxin activity. The hybrid molecule, originally called EGFATF-PE, and later EGFR-targeted bispecific angiotoxin (eBAT), consists of human EGF, human ATF and a modified Pseudomonas (PE) toxin which induces tumor killing at pico-molar concentrations [176]. Several studies report the capability of eBAT to counteract tumor growth and regulate tumor microvasculature in solid tumors. eBAT is cytotoxic on glioblastoma cells In vitro, and reduces human glioblastoma tumor size in nude rats [176,177,178]. Additionally, eBAT has been shown to exert cytotoxicity on head and neck squamous cell carcinoma and triple-negative breast cancer cells, both overexpressing EGFR and uPAR on the cell surface, and inhibit tumor growth when neck squamous cell carcinomas were implanted in nude mice [179]. Recently, eBAT-based therapy has been proposed for the treatment of sarcomas, a rare and aggressive mesenchymal malignancy often recurring despite the improvement of the local therapies [180]. eBAT was tested on RH30, a human sarcoma cell line, expressing both uPAR and EGFR, and on a TC-71 Ewing sarcoma cell line, expressing only uPAR. eBAT toxin-induced killing ability was evaluated, both in in vitro assays and in an in vivo murine model. In both RH30 and TC-71 tumor-bearing mice, the treatment reduced the growth of tumors. The ability to simultaneously target both the receptors was very effective for the complete tumor regression in an animal model [181,182]. Moreover, eBAT has been tested and found to be cytotoxic against hemangiosarcoma and rhabdomyosarcoma and their stem cell population, as well as against human osteosarcoma and ovarian adenocarcinoma [183,184]. Trials in dogs have been accomplished with good results, encouraging the idea to plan future clinical studies in humans [185,186].

## 9. Monoclonal Antibodies Inhibiting uPA/uPAR Functions

A polyclonal antibody recognizing the uPAR84–95 sequence abolishes uPAR binding to FPR1 and prevents uPA-dependent cell adhesion and migration [92,187,188], the chemotaxis of melanoma cells and the invasion of matrices and endothelial monolayers [76]. Furthermore, monoclonal antibodies against different epitopes of uPAR and uPA have been generated. Among these, the humanized huATN-658, anti-uPAR antibody was shown to exert anti-proliferative effects on prostate tumor cells implanted in murine models [189,190]. The huATN-658 binds to uPAR even if uPAR is bound to uPA since it recognizes an epitope located on the C-terminal sequence of the DIII uPAR domain, which are located the uPAR binding regions for CD11b integrin [191]. Recently, Niaz Mahmood has demonstrated that the combination of huATN-658 + Zometa synergistically inhibits the proliferation and invasion of human MDA-MB-231 cells in vitro and reduces the osteoclastic activity and number of skeletal lesions by human breast cancer in murine models [192]. 

Mazar and coworkers identified ATN-291, a specie-specific IgG1 monoclonal antibody which recognizes the kringle domain of human uPA with a Kd ~0.5 nM and is internalized in a uPA-specific manner [193]. ATN-291 was conjugated to nanobins encapsulating either arsenic trioxide packaged with cisplatin [NB(Pt,As)] or doxorubicin [PCN(DXR) [163,194]. The binding to uPA increased the uptake of ATN-291-NB/PCN, favoring its internalization into tumors expressing either uPA or uPAR. The nanobins containing chemotherapeutic drugs enhance the efficacy and the tolerability of the antitumor compounds [162,163]. Recently, starting from the ATN-291 antibody, Weifei Lu and collaborators, developed the ATN-291 F(ab′)2 which retained the selectivity and specificity for uPA in MDA-MB-231 breast cancer cells, compared with the entire ATN-291. Subsequently, ATN-291 F(ab′)2 was conjugated with NOTA-Bn-NCS for PET imaging. The monitoring of radiolabeled ATN-291 F(ab′)2 permits to follow the changes of uPA levels all along the treatment [195]. 

## 10. Discussion and Conclusions

The urokinase and its receptor regulate cancer initiation, progression and metastasis. Their high levels in tumor tissues, blood serum or organic fluids do correlate with a poor prognosis in cancer patients. As largely described in the previous sections, a significant effort has been made to design and develop compounds affecting the uPA/uPAR system with the goal to inhibit human cancer progression. Many compounds that inhibit this system, by blocking or reducing metastasis formation, have been investigated both in vitro and in animal models. Unfortunately, even though some of them are very effective in preclinical studies and animal models [12], few clinical trials including uPAR inhibitors has been accomplished, until now, whereas only a few clinical studies have been conducted, using uPA inhibitors (Table 1). It is noteworthy that the multiple interactions between uPA/uPAR and the factors constituting the so called “interactome” make it difficult to completely block the system [196]. A further problem is represented by the specie-specificity and structural flexibility of uPAR that makes it difficult to obtain agents able to completely block all uPA and uPAR interactions. Regarding specie-specificity, it is worthy to remember that the binding affinity of human-soluble uPAR for murine uPA is 68-fold weaker than that for the human uPA, and that the binding of mu-uPAR for human uPA is 46-fold weaker [197]. These differences could affect the pharmacokinetic properties of drugs and alter the data obtained for human uPAR inhibitors in mouse models. Additionally, it has to be taken into account that the efficacy of human uPAR inhibitors is commonly validated in mouse model systems or monkeys [198]. To overcome these difficulties, Cai Yuan and colleagues made the suggestion to “design inhibitors that cross species barriers according to the specie-specific residues of uPA (Asn22 and Trp30 in humans), thereby blocking uPAR/uPA and uPAR/Vn interactions simultaneously”. It is to be hoped that, in the near future, other clinical trials will be conducted to evaluate the role of the inhibitors of the uPA–uPAR system as diagnostic and therapeutic drugs.

## Figures and Tables

**Figure 1 cancers-14-00498-f001:**
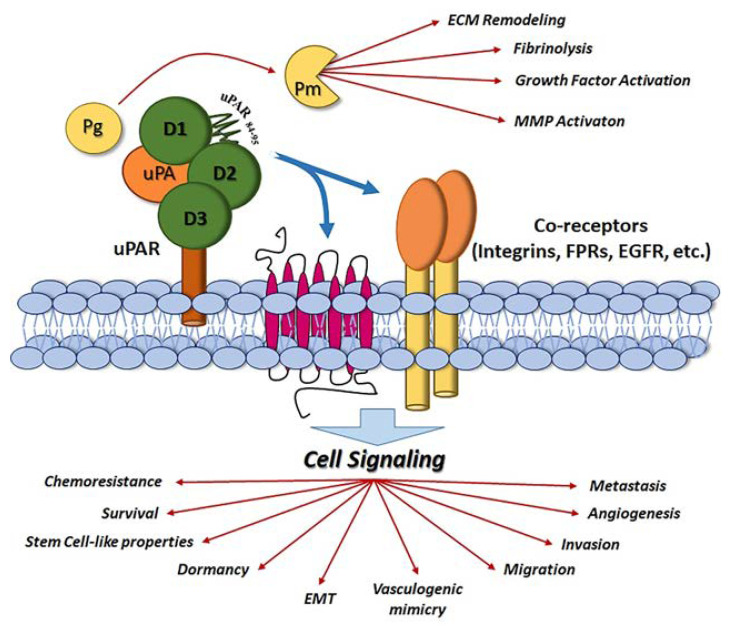
Schematic representation of uPA/uPAR system at cell surface. uPA binding to uPAR promotes plasminogen (Pg) activation to plasmin (Pm), which is responsible for catalytic as well as other activities in the cell nearby. uPAR co-receptors and uPAR modulated cell signaling in cancer cells are indicated (see Section 2 and Section 3 for details).

**Table 1 cancers-14-00498-t001:** Clinical trials with uPA/uPAR system inhibitors.

Compounds	Mechanism of Action	Study (Registration Number, ClinicalTrials.gov)	Phase	Endpoints
WX-UK1	Inhibition of uPA catalytic activity	Urokinase-plasminogen activator (uPA) Inhibitor WX-UK1 in combination with capecitabine in advanced malignancies (NCT00083525).	I	Safety, tolerability, pharmacokinetics and pharmacodynamics.
WX-671	Inhibition of uPA catalytic activity	Combination of oral WX-671 plus capecitabine vs. capecitabine monotherapy in first-line HER2-negative metastatic breast cancer (NCT00615940).	II	Clinical efficacy in combination or in monotherapy.
WX-671	Inhibition of uPA catalytic activity	Gemcitabine with or without WX-671 in treating patients with locally advanced pancreatic cancer that cannot be removed by surgery (NCT00499265).	II	Clinical efficacy in combination or in monotherapy.
Å6	Inhibition of uPAR/suPAR binding to uPA	Efficacy of Å6 in ovarian cancer patients following first-line chemotherapy and a rising CA125 levels (NCT00083928).	II	Safety and clinical efficacy.
Å6	Inhibition of uPAR/suPAR binding to uPA	A6 in treating patients with persistent or recurrent ovarian epithelial cancer, fallopian tube cancer or primary peritoneal cancer (NCT00939809).	II	Clinical efficacy.
Å6	Inhibition of uPAR/suPAR binding to uPA	Safety, tolerability and efficacy of A6 in patients with chronic lymphocytic leukemia (CLL) (NCT02046928).	II	Safety, tolerability and clinical efficacy.
68Ga-NOTA-AE105	Antagonization of uPA/uPAR interaction	uPAR PET for prognostication in patients with non-small cell lung cancer, malignant pleural mesothelioma and large cell neuroendocrine carcinoma of the lung (NCT02755675).	II	Efficacy as a prognostic tool.
68Ga-NOTA-AE105	Antagonization of uPA/uPAR interaction	uPAR PET/MRI in patients with prostate cancer for evaluation of tumor aggressiveness (NCT03307460).	II	Efficacy as a prognostic tool.
68Ga-NOTA-AE105	Antagonization of uPA/uPAR interaction	uPAR PET/CT in radium-223-dichloride treatment of patients with metastatic castration-resistant prostate Cancer (NCT02964988).	II	Efficacy as a prognostic tool.
68Ga-NOTA-AE105	Antagonization of uPA/uPAR interaction	uPAR PET/CT for preoperative staging of breast cancer patient (NCT02681640).	II	Efficacy as a prognostic tool.
68Ga-NOTA-AE105	Antagonization of uPA/uPAR interaction	uPAR PET/MRI in glioblastoma multiforme (NCT02945826).	II	Efficacy as a diagnostic and prognostic tool.
68Ga-NOTA-AE105	Antagonization of uPA/uPAR interaction	Phase II trial: uPAR PET/CT for prognostication in head cancer and neck cancer (NCT02965001).	II	Efficacy as a prognostic tool.
68Ga-NOTA-AE105	Antagonization of uPA/uPAR interaction	uPAR PET/CT and FDG PET/MRI for preoperative staging of bladder cancer (NCT02805608).	II	Efficacy as a diagnostic and prognostic tool.
68Ga-NOTA-AE105	Antagonization of uPA/uPAR interaction	PET/CT imaging of uPAR-expression in patients with neuroendocrine tumors using 68Ga-NOTA-AE105 (NCT03278275).	II	Efficacy as a prognostic tool.

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
