# Peer review of "Therapeutic Strategies Targeting Urokinase and Its Receptor in Cancer"

_cancers, 2022, doi:10.3390/cancers14030498_

Round 1

Reviewer 1 Report

A comprehensive review of the role of urokinase and targets of the urokinase pathway in development with potential clinical application in cancer management. I would however recommend that the review is revised in terms of the English language. 

Author Response

Thank you very much for your suggestions and nice revision of the manuscript  entitled “Therapeutic Strategies Targeting Urokinase and its Receptor in Cancer”   by Maria Teresa Masucci et al.

As you proposed, the review has been revised in term of English language. All the corrections are marked in red letters, so to be easily identified.

Kind regards. Sincerely yours

Maria Teresa Masucci, MD, PhD

Reviewer 2 Report

The review by MT Masucci et.al, entitled “Therapeutic Strategies Targeting Urokinasae and its Receptor in Cancer”, is a valuable work encompassing many of the main approaches aimed at the specific inhibition of uPA and uPAR in cancer therapy. The work is written in a clear and exhaustive way and provides the main theoretical foundations necessary for understanding the methods described. However, this reviewer believes that the work can be further enriched and completed by the inclusion of a chapter that has been completely disregarded by the authors: the inhibition of uPAR expression through “gene therapy”. There are numerous contributions in this regard, of which we provide an example, which is not necessarily complete. For example, Cancer Res. 1995 Jan 1; 55 (1): 90-5: in this work the authors demonstrate that uPAR aODN (anti-messenger oligodeoxynucleotides) have a potent inhibition of invasiveness of malignant cells in vitro. The anti-u-PAR ODN (5'-CGG-CGG GTG ACC CAT GTC-3 '), was designed to encompass the translation start site of the targeted mRNA, corresponding to residues 44-61 of the human u-PAR gene cDNA sequence. The in vivo effects have been demonstrated in models of melanoma (Int J Cancer. 2004; 110 (1): 125-33), of bone metastases of prostate cancer (Gene Ther. 2005; 12: 702-14), and of human hepatocellular carcinoma cells (Cancer Gene Ther. 2003; 10: 112-20). This type of approach has been discussed by Liliana Ossowski (Effect of antisense inhibition of Urokinase receptor on malignancy. Ossowski L. Curr Top Microbiol Immunol. 1996; 213 (Pt 3): 101-12). In addition to this limitation, the present review does not consider in detail the findings on the control of uPAR in the inhibition of angiogenesis, a subject on which there is a vast literature in both oncology and inflammation. This reviewer is sure that the inclusion of the aforementioned topics in this beautiful and accurate review can increase the completeness of the work and give reason to the researchers who have engaged in these lines of research.

Author Response

Thank you for your very precise suggestions and revision  of the manuscript  entitled “Therapeutic Strategies Targeting Urokinase and its Receptor in Cancer”   by Maria Teresa Masucci et al.,  which I and my colleagues appreciated very much.

As you suggested, a new Paragraph, numbered n° 7, entitled  “Genetic inhibitors of uPA and uPAR expression”, describing the inhibition of uPA and uPAR expression through genetic inhibitors,  and the papers you proposed have been included, plus some other contributions.

Also, the role of uPAR in angiogenesis has been discussed in more detail, and inserted as a part of the Paragraph n° 3 “The Urokinase Plasminogen  Activator Receptor”, starting from line n° 185.

The two added sections are written in red letters, so to be easily identified.

Corrections of English language and syntax are also in red letters.

Kind regards. Sincerely yours

Maria Teresa Masucci, MD, PhD

Reviewer 3 Report

The objective of the authors was to provide review of current literature about uPA/uPAR structure and functional relationships and their implication in various processes of carcinogenesis. The authors have discussed about therapeutic implications and the findings from several studies targeting uPA-uPAR system in different cancers. The authors have provided nice overview on the inhibitors targeting uPA-uPAR system. The authors have also summarized compounds tested in different phases of clinical trials targeting the uPA-uPAR axis. It is very informative review article but the manuscript could not be considered for publication in the current form for following reasons.

Major concerns:

1. The authors are requested to rewrite simple summary. The authors intended message is lost with sentence structure and syntax errors. 
2. The authors are also recommended to use English editing services from the MDPI or other authorized English editing service providers. There are several errors in the abstract and across manuscript. The authors are requested to proofread the entire manuscript and fix the issues with assistance from English editing services.

Author Response

Thank you very much for your revision, comments and suggestions of the manuscript  entitled “Therapeutic Strategies Targeting Urokinase and its Receptor in Cancer”   by Maria Teresa Masucci et al.,  

As you suggested, the Simple Summary has been rewritten.

Moreover, the English language has been revised and corrected.

Either the new Simple Summary and all the corrections are in red letter to be easily identified.

Kind regards. Sincerely yours

Maria Teresa Masucci, MD, PhD

Round 2

Reviewer 2 Report

Authors have accurately considered all the main points raised by this reviewer. Now this excdellent review may be published in its present form.

Reviewer 3 Report

The authors have addressed the concerns and the manuscript could be considered for publication.